# Dissemination of antibiotic resistance genes from antibiotic producers to pathogens

Xinglin Jiang[1], Mostafa M. Hashim Ellabaan[1], Pep Charusanti[1], Christian Munck[1], Kai Blin[1], Yaojun Tong[1], Tilmann Weber[1], Morten O. A. Sommer[1] & Sang Yup Lee[1,2]

It has been hypothesized that some antibiotic resistance genes (ARGs) found in pathogenic bacteria derive from antibiotic-producing actinobacteria. Here we provide bioinformatic and experimental evidence supporting this hypothesis. We identify genes in proteobacteria, including some pathogens, that appear to be closely related to actinobacterial ARGs known to confer resistance against clinically important antibiotics. Furthermore, we identify two potential examples of recent horizontal transfer of actinobacterial ARGs to proteobacterial pathogens. Based on this bioinformatic evidence, we propose and experimentally test a 'carry-back' mechanism for the transfer, involving conjugative transfer of a carrier sequence from proteobacteria to actinobacteria, recombination of the carrier sequence with the actinobacterial ARG, followed by natural transformation of proteobacteria with the carrier-sandwiched ARG. Our results support the existence of ancient and, possibly, recent transfers of ARGs from antibiotic-producing actinobacteria to proteobacteria, and provide evidence for a defined mechanism.

---

[1] The Novo Nordisk Foundation Center for Biosustainability, Technical University of Denmark, Kemitorvet Bygning 220, 2800 Kgs. Lyngby, Denmark. [2] Metabolic and Biomolecular Engineering National Research Laboratory, Department of Chemical and Biomolecular Engineering (BK21 Plus Program), Center for Systems and Synthetic Biotechnology, Institute for the BioCentury, Korea Advanced Institute of Science and Technology (KAIST), Daejeon 34141, Republic of Korea. Correspondence and requests for materials should be addressed to T.W. (email: tiwe@biosustain.dtu.dk) or to M.O.A.S. (email: msom@bio.dtu.dk) or to S.Y.L. (email: leesy@kaist.ac.kr).

The discovery of antibiotics from microorganisms and their development into clinical drugs constitutes one of the greatest advances in medical history, but the acquisition and dissemination of genes that confer antibiotic resistance among pathogens has severely curtailed the effectiveness of many of these compounds. Elucidating the origins of these antibiotic resistance genes (ARGs) and the mechanisms mediating their spread to pathogens is consequently a public health priority[1]. Actinobacteria, especially of the genus *Streptomyces*, produce many clinically important antibiotics. In most cases, the gene clusters encoding the biosynthesis of these compounds also contain resistance genes as a self-protecting mechanism towards these compounds[2] or to modulate their signalling activity[3]. As early as 1973, it was hypothesized that the enzymes found in Gram-negative pathogens that inactivate aminoglycoside antibiotics could have originated from the ARGs of actinobacteria that produce this class of antibiotics through ancient horizontal gene transfer (HGT), based on the discovery that they employ the same enzymatic mechanisms[4]. This 'producer hypothesis' was subsequently proposed for additional ARGs, for example, some class-A β-lactamases[5] in Gram-negative pathogens, and erythromycin[6] and vancomycin resistance genes[7] in Gram-positive pathogens. However, in these cases the sequence similarities between ARG proteins in actinobacteria and Gram-negative pathogens are low, making it difficult to distinguish if they result from ancient HGT or from other types of evolutionary processes[8].

Compared with ancient transfers, recent ARG dissemination from actinobacteria to pathogens may pose an even more urgent threat to human health[9], as actinobacterial ARGs make up a large portion of the environmental resistome. In addition to self-protecting ARGs, most actinobacteria also carry ARGs horizontally obtained from other actinobacteria[6,10]. In an investigation of actinomycetes from soil, isolates were on average resistant towards seven to eight antibiotics from a collection of 21 representative antibiotics[11]. In a functional metagenomics study using *Escherichia coli* as an expression host, actinobacteria were found to be the most enriched source of resistance-conferring DNA

fragments relative to their abundance[12]. However, since no recent transfer from actinobacteria to Gram-negative pathogens has been discovered and many recent studies showed that phylogenetic and ecological boundaries are two major ARG transfer barriers[12–15], the clinical relevance of this large resistome remains unclear[16,17].

Here, we examine a large collection of known *Streptomyces* ARGs, and find more examples supporting the 'producer hypothesis'. Specifically, we provide evidence that two ARGs conferring resistance against chloramphenicol and lincomycin might have been recently transferred from actinobacteria to human and animal pathogens. Based on their surrounding sequence, we propose and test a potential mechanism mediating gene transfer from Gram-positive actinobacteria to Gram-negative pathogens.

## Results

**Actinobacterial ARGs have related proteins in proteobacteria.** Proteobacteria encompass many important human pathogens such as *Escherichia coli*, *Pseudomonas aeruginosa* and others. We therefore sought to examine the similarity between corresponding ARGs in actinobacteria and proteobacteria. First, we extracted all experimentally validated *Streptomyces* ARG proteins from ARDB[18] and CARD[19] databases. A majority (39 of 57) of them are reported to have self-protecting roles or are located within antibiotic biosynthesis gene clusters as analysed by antiSMASH[20] (Supplementary Data 1). Their most similar homologues in proteobacteria were subsequently identified from the NCBI non-redundant protein database by BLASTP, with sequence identities ranging from 23 to 68%. Seven of these proteobacterial proteins have sequences more similar to actinobacterial proteins than to proteins from any other phyla, including proteobacteria (Supplementary Data 1). Furthermore, phylogenetic trees were constructed and showed that 12 (including the above seven) of the proteobacterial proteins may have originated from actinobacteria by interphylum HGT (Fig. 1 and Supplementary Fig. 1)[21]. Potential HGTs in the opposite direction, from proteobacteria to actinobacteria, were also noticed in some

| Streptomyces ARG proteins | | | | Proteobacteria proteins | |
|---|---|---|---|---|---|
| ARG | Antibiotic | Resistance mechanism | Sequence identy | Protein ID | The host strain is isolated from |
| *cml_e* | Chloramphenicol | Efflux | - - - - - 63% - - - - - ➤ | WP_005297378.1 (*cmx*) | Patients |
| *lmrA* | Lincomycin | Efflux | - - - - - 50% - - - - - ➤ | WP_038989331.1 (*lmrA*) | Farm animals |
| *aph33ia* | Streptomycin | Inactivating enzyme | - - - - - 51% - - - - - ➤ | WP_031942890.1 (*aph(3")*) | Patients |
| *otrb* | Tetracycline | Efflux | - - - - - 39% - - - - - ➤ | WP_048022769.1 | Cabbage (98% to WP_0.64384801.1 from clinical isolates of *Klebsiella pneumoniae*)[a] |
| *cata5* | Chloramphenicol | Inactivating enzyme | - - - - - 56% - - - - - ➤ | WP_053238935.1 | Soil |
| *aph33ia* | Streptomycin | Inactivating enzyme | - - - - - 51% - - - - - ➤ | WP_037160408.1 | Populus root |
| *tet* | Tetracycline | Target protection | - - - - - 48% - - - - - ➤ | WP_046110059.1 | Soil |
| *rph* | Rifamycin | Inactivating enzyme | - - - - - 68% - - - - - ➤ | WP_014395981.1 | Soil |
| *pur8* | Puromycin | Efflux | - - - - - 48% - - - - - ➤ | WP_043284319.1 | Soil |
| *pac* | Puromycin | Inactivating enzyme | - - - - - 47% - - - - - ➤ | WP_046974149.1 | Entomopathogenic nematodes |
| *tcma* | Tetracenomycin_c | Efflux | - - - - - 35% - - - - - ➤ | EFG83002.1 | Environmental |
| *facT* | Factumycin | Efflux | - - - - - 43% - - - - - ➤ | WP_045683650.1 | Marine plant |
| *sul1* | Sulfonamide | Target replacement | ◄ - - - - - 95% - - - - - | ALJ92876.1 (*sul1*) | Patients |

**Figure 1 | Proteobacterial proteins related to *Streptomyces* ARG proteins.** Relationships between all experimentally validated *Streptomyces* ARG proteins and their most similar homologues in proteobacteria were analysed by phylogenetic analysis (Supplementary Fig. 1). Selected pairs that might be connected via interphylum HGT are summarized here, with the proposed transfer direction indicated by an arrow. Protein sequence identity was determined by BLASTP. [a]The proteins were BLASTed against pathogen genome database PATRIC to see if they have close homologues (sequence identity threshold of 90%) in pathogens.

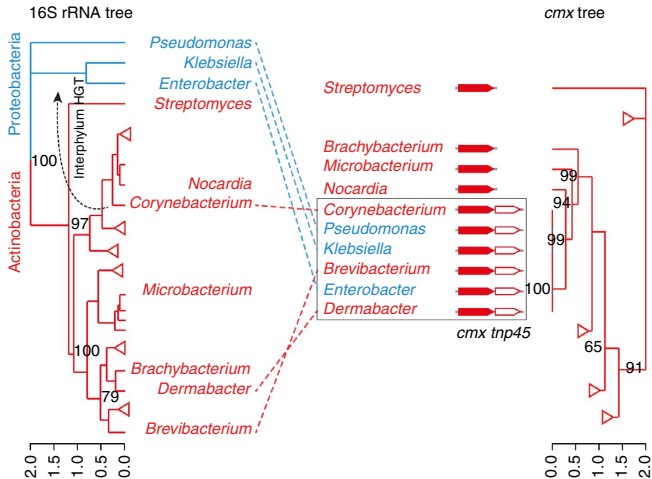

**Figure 2 | A tanglegram between the phylogeny of the genus representative hits of *cmx* and the host phylogeny of their corresponding 16S rRNA sequences.** The red labels correspond to Gram-positive actinobacteria, while the blue labels correspond to Gram-negative proteobacteria. Genomes were selected if they have BLASTP hits against the Cmx protein (WP_005297378.1). The best hit per genus is taken as the representative hit. Hits with over 99% identity to WP_005297378.1 are framed. These *cmx* genes are located in transposons together with a transposase gene *tnp45*. Triangles represent collapsed branches. The full trees are provided in Supplementary Fig. 4.

trees. In particular, the *Streptomyces* ARG *sul1* (AFN41071.1) shares 95% sequence identity with a proteobacterial gene (Supplementary Fig. 1).

By examining neighbouring genes, we also found evidence for potential cotransfer of additional genes in the case of the ARG *pac* (conferring resistance to puromycin). The products of six neighbouring genes also show higher similarity to actinobacterial proteins than to proteins from any other phyla, suggesting that they might have been transferred together from actinobacteria to proteobacteria (Supplementary Fig. 2).

Of the 12 proteobacterial proteins that might have originated from actinobacteria, nine are encoded in genomes of environmental species, one of which has close homologues encoded in the genomes of pathogens from the database PATRIC[22]. The other three proteins are harboured by pathogens, including the well-known aminoglycoside-inactivating phosphotransferase APH(3″) (WP_031942890.1)[4].

**Recent ARG transfers from actinobacteria to proteobacteria.** By analysing DNA sequence composition signatures using RAIphy[23], we found that two of the proteobacterial proteins, chloramphenicol exporter Cmx (WP_005297378.1) and lincomycin exporter LmrA (WP_038989331.1), may have been recently transferred from actinobacteria, since the proteobacterial genes still retain actinobacterial sequence signatures (Supplementary Data 2)[8]. Cmx is found in clinical isolates of *P. aeruginosa*, *Klebsiella oxytoca* and *Enterobacter asburiae* (Supplementary Data 3). It is 63% identical (100% coverage) to the chloramphenicol resistance protein (P31141) from *Streptomyces lividans* 66 (ref. 24), and 52% identical (99% coverage) to the self-protecting resistance protein (WP_015032122.1) from chloramphenicol producer *Streptomyces venezuelae* (Supplementary Fig. 3). Furthermore, its gene was found to be identical or almost identical (identity over 99%) to genes from many non-*Streptomyces* actinobacteria (Fig. 2), further supporting that the interphylum gene transfer happened recently.

**Possible mechanisms of the ARG transfers.** Conjugation or transduction from actinobacteria to proteobacteria has so far not been described. Many proteobacteria can take up free DNA through natural transformation, but because of low sequence similarity, actinobacterial DNA has a low probability of being incorporated into proteobacterial genomes by homologous recombination[25]. In theory, the DNA can be inserted randomly by non-homologous end joining, but the frequency is extremely low[26]. Transposase/integrase-mediated recombination after natural transformation is possible if a transposase or integrase is encoded in the DNA and can be expressed before the DNA is degraded[27]. We noticed that *cmx* is colocalized with an actinobacterial transposase gene *tnp45* forming a transposon[28], and the intact transposon can be found in both actinobacteria and proteobacteria (Fig. 2). Initially, we hypothesized that a DNA fragment containing the transposon released from dead actinobacteria could be taken up by proteobacteria, and the transposon could insert itself into the new genome by the activity of its transposase. To test this hypothesis, we examined if the transposition activity, which has been experimentally validated in *Corynebacterium glutamicum*[29], was also functional in proteobacteria. However, no transposition was detected in our experimental set-up (details are described in Methods section).

However, by further analysing the DNA sequence flanking the *cmx* transposon in both phyla, we arrived at a different hypothesis to explain the interphylum HGT, which we call the 'carry-back' model (Fig. 3a). First, a proteobacterial sequence is transferred from proteobacteria into actinobacteria by conjugation, a mechanism known to be highly efficient[30]. Next, it recombines with actinobacterial DNA, for example, by the transposition of *cmx* transposon, forming a sandwich structure of actinobacterial DNA flanked by proteobacterial DNA. Then, the sandwich structured DNA released from dead actinobacterial cells is taken up by nearby proteobacteria through natural transformation and incorporated via homologous recombination. In support of this 'carry-back' model, we identified DNA sequences representing all the proposed intermediates (Fig. 3b and Supplementary Data 4). The carrier sequence is a fragment from class 1 integron In4, composed of an IS6100 insertion sequence, an acyltransferase gene (*orf5*) and *sul1* (ref. 31). This sequence is widely distributed among proteobacteria, and frequently found on conjugative plasmids (Fig. 3b). The same carrier sequences can be also found in actinobacteria, for example, in *Corynebacterium diphtheriae* BH8. Part of the carrier sequence, the *sul1* gene, can be found in *Streptomyces* sp. 1AL4 (ref. 32). The sandwich structure with *cmx* transposon inserted between IS6100 and *orf5* is found in *C. diphtheriae* BH8 (Fig. 3b) and *Corynebacterium resistens* plasmid pJA144188 (Fig. 3c), although with additional resistance genes and mobile elements inserted as well. The final product of the interphylum transfer, the sandwich structure in proteobacterial genome, is found in *E. asburiae* 35642 and *K. oxytoca* CHS143 (Fig. 3b). All these intermediates fit the proposed 'carry-back' model, and appear to be stably maintained in their respective hosts, thus potentially providing sufficient time for the multistep process to be accomplished.

The 'carry-back' model is also supported by the second case of potential recent transfer that we have identified in the *in silico* studies. LmrA (WP_038989331.1), found in *Salmonella enterica* and *Escherichia coli* isolated from farm animals, is 50% identical to the self-protecting lincomycin pump CAA42550 from *Streptomyces lincolnensis*. Its gene and an adjacent regulator gene still retain actinobacterial sequence signatures (Supplementary Data 3), suggesting that they may have been transferred from actinobacteria recently. Interestingly, they are located in an RSF1010-like plasmid (Supplementary Fig. 5), which could have

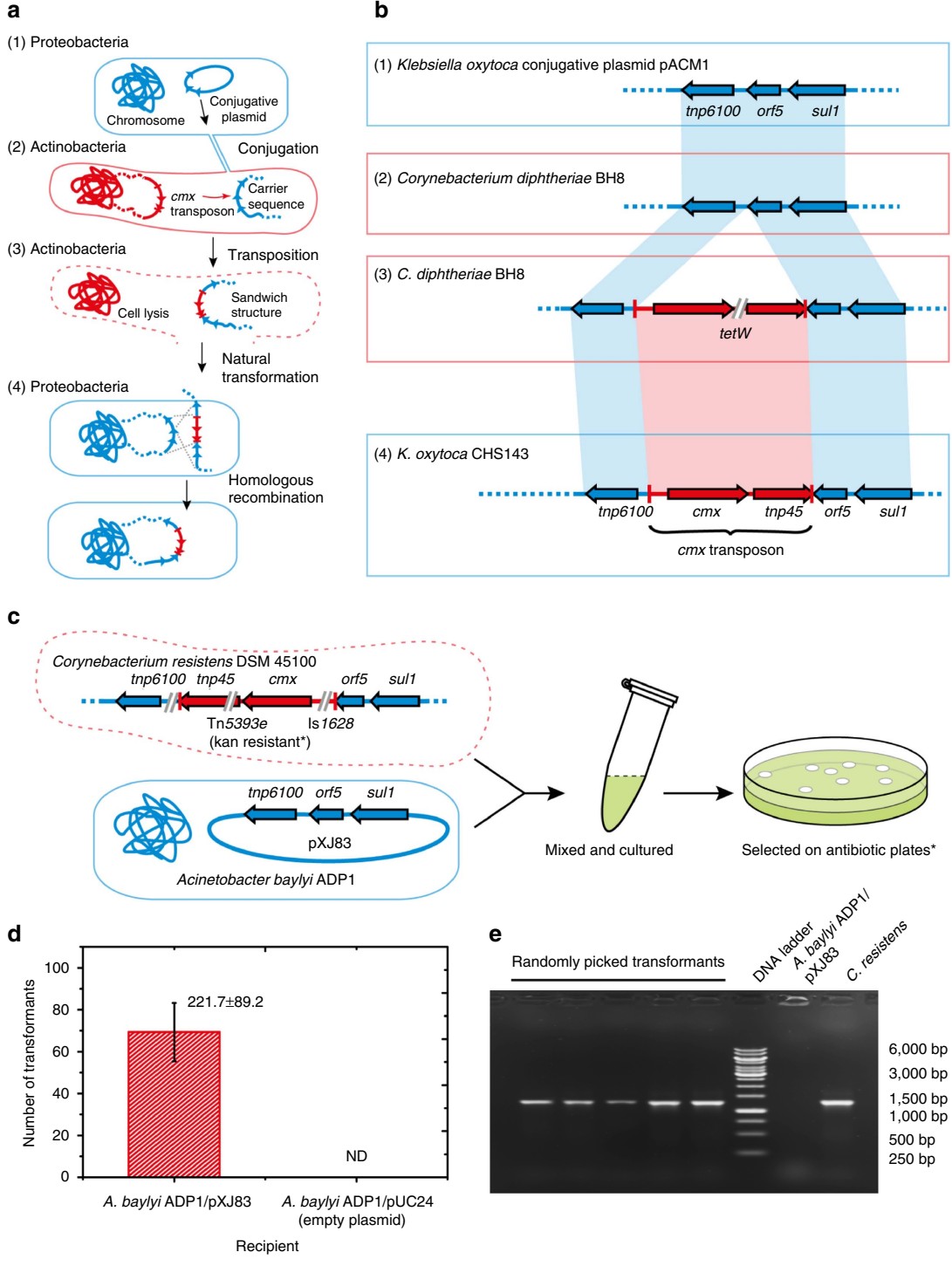

**Figure 3 | Transfer of a resistance gene from actinobacteria to proteobacteria by the 'carry-back' model.** Actinobacterial and proteobacterial cells are marked in red and blue, respectively. DNA sequences of actinobacterial and proteobacterial origins are likewise labelled in red and blue. Dashed line represents DNA that can be either a fragment of a chromosome or a plasmid. The terminal inverted repeats of *cmx* transposon are represented by red bars. (**a**) The proposed 'carry-back' interphylum gene transfer mechanism. (**b**) Examples of actual sequences found in different bacteria that resemble the proposed intermediates: (1) carrier sequence in proteobacteria; (2) carrier sequence in actinobacteria; (3) sandwich structure with *cmx* transposon flanked by carrier sequences in actinobacteria; (4) *cmx* transposon incorporated into genome in proteobacteria. Double-slash indicates additional resistance genes and mobile elements inserted into the proposed intermediate structures. NCBI accession numbers of these intermediate sequences are provided in Supplementary Data 4. (**c**) Experimental reconstruction of the interphylum gene transfer of *cmx* from actinobacteria to proteobacteria. *C. resistens* DSM 45100 naturally carries the sandwich structure. Plasmid pXJ83 is constructed by cloning the carrier sequence from *P. aeruginosa* BM4530 onto pUCP24 vector. *Kanamycin (kan) instead of chloramphenicol was used for the selection plates. Kanamycin resistance is coded by Tn5393e inserted in the *cmx* transposon. (**d**) Efficiency of natural transformation using *A. baylyi* ADP1/pXJ83 as recipient strain (error bar: s.d.; $n = 3$). *A. baylyi* ADP1/pUCP24 was used as a negative control. The detection limit is one transformant per $10^9$ colony-forming unit (c.f.u.) of recipient cells. ND, not detected. (**e**) Colony PCR of transformants using primers that target *cmx*. *C. resistens* DSM 45100 was used as a positive control. *A. baylyi* ADP1/pXJ83 was used as a negative control.

served as the carrier sequence as the ability of RSF1010 (NC_001740.1) to be mobilized by conjugation from *E.coli* into *Streptomyces* and *Mycobacterium* has been documented[33].

**Experimental reconstruction of the interphylum transfer of *cmx*.** To provide further evidence for the hypothesis, we reproduced the interphylum gene transfer of *cmx* experimentally. *C. resistens* DSM 45100, which naturally has the sandwich structure sequence, was used as the actinobacterial *cmx* donor. *Acinetobacter baylyi* ADP1, a model strain for HGT studies[27,34] (Fig. 3c) with a cloned IS6100-orf5-sulI sequence, was used as the *cmx* acceptor to mimic proteobacterial pathogens with the same sequence, such as *Acinetobacter baumannii* D4 (Supplementary Data 4). Heat-killed and lysed *C. resistens* cells were added into *A. baylyi* ADP1 culture, incubated and followed by selection on kanamycin plate (kanamycin resistance was coded by the Tn5393e inserted between *cmx* and *tnp45* in *C. resistens*, see Fig. 3c). The obtained colonies showed no chloramphenicol resistance, probably because the promoter activity was disrupted by the IS1628 inserted in front of *cmx* (Fig. 3c). However, the existence of *cmx* in these colonies was confirmed by colony PCR and sequencing (Fig. 3e), demonstrating that the *cmx* gene was transferred to *A. baylyi*. The negative control, an *A. baylyi* ADP1 strain without IS6100-orf5-sulI sequence, did not generate any colonies after the same treatment (Fig. 3d), confirming the involvement of the carrier sequence in the ARG transfer.

To confirm the resistance activity of *cmx* in proteobacteria, we cloned an intact *cmx* transposon free of IS1628 into *E. coli* and observed increased chloramphenicol tolerance when promoters were provided (Supplementary Fig. 6)[29].

**Possible dissemination route of *cmx*.** Cmx family proteins are widely spread among *Streptomyces* and other actinobacteria. Its recruitment into *tnp45* transposon may have occurred in soil environment, as both free *cmx* and transposon-carried *cmx* have been found in soil actinobacteria. For example, *C. glutamicum* 1014 contains *cmx* on the *tnp45* transposon, whereas *Arthrobacter* sp. 161MFSha2.1 contains *cmx* but not *tnp45*. The two *cmx* genes are 93% identical (Supplementary Fig. 7). We hypothesize that the *cmx* transposon found in commensal and pathogenic actinobacteria was probably obtained from the related strains from soil. Then, the *cmx* transposon might have been transferred into Gram-negative pathogens by the 'carry-back' mechanism. Currently, *cmx* transposon-harbouring pathogens have been isolated in Asia, Europe, United States of America and especially South America (Supplementary Data 3).

## Discussion

Clinical ARGs have complex and diversified origins[6,16], and our results indicate that some of them might have originated from actinobacteria. Ancient ARG transfers are proposed to have occurred in soil environments and then passed on to pathogens in modern times[35]. Recent transfers from actinobacteria to pathogens might also be possible, as suggested by the examples of *cmx* and *lmrA*.

Proteobacteria are well known to be able to transfer DNA to organisms from other phyla and even other kingdoms by conjugation[34]. A recent study suggested that conjugation from proteobacteria to actinobacteria might happen frequently in soil[30]. Thus, the 'carry-back' mechanism might have mediated the HGTs from actinobacteria to proteobacteria in soil using conjugative plasmids as the carrier sequence. In modern times, likely caused by increased selection pressure due to the extensive use of antibiotics, mobile genetic elements including conjugative plasmids, integrons and transposons tend to be clustered together with ARGs forming mobile multidrug-resistant units. These units showed extraordinary capability of spreading among commensals, pathogens and even environmental bacteria in water and soil[36]. Not surprisingly, the *cmx* carrier sequence was from a widespread conjugative class I integron and is composed of the IS6100 transposon and another resistance gene *sul1*. Accordingly, the *cmx* carried back was incorporated into such units, facilitating further dissemination.

In addition to being a gene transfer barrier, different phyla also mean distinct cell environments for gene expression, regulation and protein function[37]. For example, due to the different cell membrane structure in proteobacteria, drug efflux pumps like Cmx and LmrA will export compounds into the periplasm instead of the extracellular environment as in actinobacteria. Future studies are required to understand how newly obtained resistance genes and their new hosts will evolve after the gene transfer.

## Methods

**Homologues of *Streptomyces* ARG proteins in proteobacteria.** All 87 ARG proteins of the genus *Streptomyces* in ARDB[18] and CARD[19] databases were downloaded (October 2016). They were further manually curated by removing artificial constructs (ARGs used as selective marker during *Streptomyces* genetic engineering) and dereplicating proteins with high sequence similarity (identity over 95% with each other). Then, they were used as queries in BLASTP analyses against the NCBI non-redundant protein database of proteobacteria to find their most similar homologues. Some of the best hits were found to be caused by sequence contamination (by the following method), that is, they were actually from actinobacteria but mislabelled as proteobacterial proteins in the database. In this case, the next best hit was analysed by the same method until a true proteobacterial protein was found. In addition, the identified proteobacterial proteins were searched (BLASTP) against the non-redundant protein database of all phyla to see if they were more similar to actinobactieral proteins than to proteins from other phyla.

**Sequence contamination check.** Because HGT studies are very sensitive to cross-contaminations in the sequence data, a manual check of possible contaminations was performed: if the best BLAST hit was from a fully assembled genome or circular plasmid sequence with experimentally determined source organism to be proteobacteria, then the hit was considered as a true proteobacterial protein. In cases where the best hit was a contig instead of a full genome, then the protein was considered as a true proteobacterial protein only if the contig was 10 kbp or larger and the contig had the best sequence match to other proteobacterial sequences in the 'nt' database. If the contig had best nucleotide matches against 'nt' from actinobacteria, it was considered an artefact caused by contamination.

**BLAST against PATRIC.** The proteobacterial proteins were BLASTed against pathogen genome database PATRIC (https://www.patricbrc.org/app/BLAST, with sequence identity threshold of 90%) to see if they have close homologues in pathogens.

**Comparison of the neighbouring area of the ARGs.** In search of cotransfer of addition genes, neighbouring area of the ARGs were compared between the *Streptomyces* genomes and proteobacterial genomes by MultiGeneBlast using amino-acid translation sequences[38].

**Bacterial genome databases for phylogenetic tree construction.** All protein data sets of bacterial genomes were downloaded from the NCBI RefSeq database available at ftp://ftp.ncbi.nlm.nih.gov/genomes/refseq/bacteria/ (October 2016). The collection covers 64,580 genomes and 55,667,859 unique proteins. To avoid contaminated genomes, we have excluded genomes that either lack the 16S rRNA gene or have multiple copies of 16S rRNA genes from multiple species.

**Construction of phylogenetic trees.** BLAST analysis for each ARG protein was performed using *blastp* command of NCBI BLAST version 2.2.31 + , with the following parameters: the maximum target sequence of 55 million sequences and E value below 1E − 50. A genome was considered if it has a hit with a minimum 30% identity and 80% coverage where coverage is defined as (number of matched amino acid + number of mismatched amino acid + gaps)/(query protein length). To simplify the phylogenetic tree, we normalized the number of hits to 100 genomes, considering the best hit per species, genus or maximum order. We then constructed phylogenetic trees for these genomes based on their 16S rRNA and constructed phylogenetic trees for the ARGs based on their protein sequences. Sequences were multiply aligned using MAFFT[39], considering

the local alignment option with default settings for the other parameters, then optimized and phylogenetic trees were generated, using the FastTree methods[40]. The generated host and ARG trees were visualized using ETE 3 package[41].

**DNA sequence signature analysis.** RAIphy[23] was used to analyse gene sequences to find their most likely taxonomic source. Relative abundance index (RAI) is calculated from the over- and underabundance statistics collected for each taxon. The set of RAI profiles was extracted from 2,773 complete bacterial genomes downloaded from NCBI ftp://ftp.ncbi.nlm.nih.gov/genomes/archive/old_genbank/Bacteria/ (March 2016). To avoid false taxonomic assignment, we excluded plasmids and genomes below 1 Mbp.

**Strains and media.** *P. aeruginosa* PA7, *C. urealyticum* DSM 7109 and *C. resistens* DSM 45100 were obtained from DSMZ and cultured in DSMZ media. *E. coli* DH5α and *A. baylyi* ATCC33305 cultures were cultured in Luria-Bertani (LB) medium. Antibiotics were added at the following concentrations: ampicillin, 50 µg ml⁻¹; kanamycin, 50 µg ml⁻¹; gentamicin, 20 µg ml⁻¹. Chloramphenicol was added at indicated concentrations.

**Cloning of *cmx* transposon and the carrier sequence.** The *cmx* transposon was PCR amplified from *P. aeruginosa* PA7 by primer xj143 (as both forward and reverse primer). Plasmid backbone was amplified from plasmid pKD46 by primer xj144 and xj145. The two PCR products were assembled by Gibson reaction, transferred into *E. coli* DH5α and selected on Amp plates. Insertion direction of *cmx* transposon was determined by sequencing with primer xj146. The resulted plasmids with *cmx* transposon in the same or opposite direction with the plasmid *repA* gene were named as pXJ79 and pXJ80, respectively. pXJ79cu and pxj80cu were constructed likewise except that the *cmx* transposon was amplified from *C. urealyticum* DSM 7109. pXJ79 was amplified with primer xj177and xj178, the product was recircularised by Gibson reaction, generating pXJ79a. By this way, promoter pLac was inserted in front of the *cmx* transposon. The carrier sequence was amplified from *P. aeruginosa* BM4530 with primer xj173 and xj174.1 and cloned on a plasmid backbone amplified from pUCP24 with primer xj171.1 and xj172, generating pXJ83. Phusion Hot Start II DNA Polymerase for PCR was from Thermo Fisher. Gibson Assembly Master Mix was from New England Biolabs. Primer sequences are listed in Supplementary Table 1.

**Test of transposition activity in *E.coli*.** *E.coli* DH5α/pXJ79cu and *E.coli* DH5α/pXJ80cu were cultured in LB containing ampicillin and chloramphenicol, respectively, at 30 °C. Subculturing was repeated five times at 1:100 dilution. After that, 0.1 ml was inoculated into 10 ml fresh LB and cultured at 42 °C overnight to eliminate the plasmids, and then spread on LB agar containing 2.5 µg ml⁻¹ chloramphenicol and culture at 37 °C until colonies appeared. The colonies were checked by colony PCR using primer xj144 and xj145 to see if they still had the plasmids. PCR-positive colonies were subjected to another round of plasmid elimination and chloramphenicol-sensitive test as described above.

**Natural transformation of *A. baylyi*.** Fifty microlitres overnight culture of *A. baylyi* ATCC33305/pXJ83 was inoculated into 1 ml fresh LB and incubated for 2 h at 120 r.p.m. *C. resistens* DSM 45100 cells from 1 ml overnight culture was washed with water, boiled for 15 min and mixed with the fresh *A. baylyi* culture. Two hundred micolitres of the mixture was spread on top of 1 ml LB agar (in 12-well plate) and cultured overnight. Then, cells were transferred to LB plates containing kanamycin. Colonies were randomly picked and checked by colony PCR using primer xj130 and xj131 for the presence of *cmx*.

**Data availability.** The authors declare that all the relevant data are provided in this published article and its Supplementary Information files, or are available from the corresponding authors on request.

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

## Acknowledgements

We thank the lab of Prof Jeff Dangl and the DOE-JGI for making their genome sequences available before publication. We thank Prof Justin Nodwell and Dr Günther Muth for discussion. We also thank Prof Patrice Courvalin for kindly providing total DNA of *P. aeruginosa* BM4530 and Prof John C. Alverdy for providing vector pucp24. M.M.H.E. acknowledges the help and assistance of Ali Syed and other members of computerome, Danish National Supercomputer for Life Sciences, in using the HPC facilities. This work was funded by the Novo Nordisk Foundation. M.O.A.S., C.M. and M.M.H.E. acknowledge funding from the Lundbeck foundation. S.Y.L. acknowledges additional funding from the Korean Ministry of Science, ICT and Future Planning (NRF-2012M1A2A2026556).

## Author contributions

X.J. and S.Y.L. conceived the project. T.W., X.J., M.O.A.S. and K.B. planned the study. X.J. executed experiments and analysed data. M.M.H.E. and X.J. performed bioinformatic analyses. All authors discussed results and wrote the manuscript together.

## Additional information

**Competing interests:** The authors declare no competing financial interests.

**Publisher's note**: 

