## [Peer Review File · Nature Communications]

Reviewers' comments:

Reviewer #1 (Remarks to the Author):

RE

The authors provide a comprehensive bioinformatics based analysis of the hypothesis that the origins of some/many ARG are different antibiotic producing microbes.. The manuscript is novel, has been well worked through, is clear and consistent. It allows reproduction of the investigation.

The results should be of high interest to others in the community. Some comments for the authors consideration that may further enhance the manuscripts impact on the thinking in the field:

Points for clarification and refinement of the discussion section

1. How do these observations fit the One Health" approach/concept?

2. What is the contribution of gene transfer from antibiotic producers to the overall diversity of transferable antibiotic resistance genes in the clinics?

It would be useful for the readers if the authors, based on their understanding of ARG databases could say something more specific about the proportion of r-genes currently known that are likely to originate from antibiotic producers. This paper is important by qualitatively demonstrating sources and a pathway, - the discussion section should include a discussion about the relative importance, from the authors point of view.

3. As pointed out by the authors, wide HGT are perhaps not, from a mechanistic point of view, likely to occur frequently. In addition, there may be other constraints to a protein 's function in a new host.

Both transfer barriers as well as barriers to functionality in new distant hosts are expected to limit stable gene transfers from antibiotic producers. The authors should comment on the latter barrier as well, in particular since efflux pumps are considered in the manuscript, - exemplifying proteins that may depend on other host membrane properties for adequate functioning and regulation.

3. Early studies by J. Davies and others showed that antibiotic preparations contained DNA from the commercial production strains - can something be said from this study about the relevance (and impact-if any) of such practices. I.e. its likely that antibiotic preparations have been contaminated with DNA from decades since the 1940s.

4. Carrier DNA can be a number of mobile genetic elements with broad host range. The need to introduce a new concept of "carriage" could be made clearer.

4. Page 6, line 126, reword "highly identical"

5. Page 11, line 230, reword "Homologous ARGs"

Kaare Magne Nielsen

Reviewer #2 (Remarks to the Author):

Nature Communications manuscript NCOMMS-16-21932

Jiang et al.

Dissemination of antibiotic resistance genes from antibiotic producers to pathogens

This manuscript presents experimental and bioinformatic evidence for the hypothesis that antibiotic resistance genes have their ultimate origins in antibiotic producing organisms. This hypothesis presupposes that antibiotics themselves might be toxic at all concentrations, and are

toxic as soon as they are synthesized within cells. Consequently producers would require resistance mechanisms. Whether this is generally the case is perhaps debatable, but it certainly might be the case in some specific circumstances. If this were the case, then the authors should refrain from the over-used 'reservoir' metaphor, since it implies the genes are somehow in storage, rather than being actively engaged in protecting the host cell.

The manuscript suffers from the somewhat scattered approach to the question, the lack of details around the data that have been assembled, and the strength of the conclusions that can be reached on the basis of these data. The first piece of evidence presented centres around the puromycin resistance gene, *pac*. The text of the paper suggests that the puromycin synthesis gene cluster and the resistance gene *pac* were transferred from *Streptomyces* into *Xenorhabdus maltophilia*, and that subsequently just the *pac* gene was transferred into *Photorhabdus temperata* (Figure 2, lines 93-96). This conclusion is not supported by the data. Yes, it is possible that this operon was transferred between Actinobacteria and Proteobacteria at some time, but the significant differences in protein identity suggest that this event happened in the deep evolutionary past. Consequently, one cannot identify the donor or the recipient of the HGT with any certainty, and since the HGT event occurred thousands, if not millions of years ago, it does not have much relevance to the current antibiotic resistance crisis. It is also certain that any number of diverse genes have been moved by HGT between phyla over the millennia, so the observation that some of these might encode antibiotic resistance should not be surprising.

The second piece of evidence is based on bioinformatic analyses of antibiotic resistance gene proteins from *Streptomyces*, recovered from the ARDB and CARD databases. These were used to search the Proteobacteria protein database for homologues. Those proteins that generated good hits were then used in further searches and to generate phylogenies (presented as supplementary data). Examining these phylogenies shows diverse patterns of potential inheritance: some genes appear to be restricted to Proteobacteria, others appear to have been acquired from Actinobacteria by HGT, others appear to have been donated to Actinobacteria from Proteobacteria, and in other cases the phylogenies suggest extensive movement to diverse phyla by HGT. So the conclusions drawn here suffer from confirmation bias. Yes, there are instances where the most parsimonious explanation is HGT from Actinobacteria, but there are also examples of every other kind of HGT movement and direction. To concentrate just on the Actinobacterial to Proteobacterial movement as if this tells us something novel is to miss the point entirely.

Finally, the authors investigate the possibility that mobile DNA elements with their origins in Proteobacteria can infiltrate the Actinobacterial genome and shuttle genetic material back to the Proteobacteria. They have convincingly demonstrated this in laboratory assays, backed up with some bioinformatic observations. Again, this is not surprising, and the focus here should be on the fitness of mosaic mobile elements whose evolutionary success lies in their ability to associate with genes that confer advantageous phenotypes. The present manuscript reads somewhat teleologically, as if this is a strategy specifically employed by the Proteobacteria.

The manuscript also requires significant copy editing, both to correct a number of typographical errors, and to improve the English expression. The methods and processes employed during the work need to be explained in more detail, and the rationale behind those procedural decisions needs to be justified.

Reviewer #3 (Remarks to the Author):

The authors purport evidence of a new mechanism for horizontal gene transfer between distantly related bacteria. Actinobacteria have long been appreciated to be the source of many of the clinically relevant antibiotics used today. As expected, and well documented in previous literature, these bacteria must also encode resistance determinant to the antibiotics they produce. As documented in many publications, this provides a wealth of antibiotic resistance determinants in

the natural environment. The current manuscript does not provide any new information in this area. A second claim in the manuscript is that a direct natural source of transfer between members of the phylum Actinobacteria and proteobacteria has been identified. However, the actual evidence reported to support this idea involves using laboratory cloning procedures to make a variety of recombination intermediates that could have occurred via transfer through various intermediate bacteria. Therefore, no new information illuminating a hereto unknown recombination step is provided in this work. In summation, the current work simply summarizes previous well established findings without providing an new mechanisms of transfer between distant important phyla.

We would like to thank the reviewers for their constructive criticism that allowed us to significantly improve our manuscript. We did our best to fully address to their comments as follows:

Reviewer #1 (Remarks to the Author):

The authors provide a comprehensive bioinformatics based analysis of the hypothesis that the origins of some/many ARG are different antibiotic producing microbes. The manuscript is novel, has been well worked through, is clear and consistent. It allows reproduction of the investigation.

The results should be of high interest to others in the community. Some comments for the authors consideration that may further enhance the manuscripts impact on the thinking in the field:

We thank the reviewer for the positive comments regarding our manuscript.

Points for clarification and refinement of the discussion section

1. How do these observations fit the One Health” approach/concept?

We have included a paragraph connecting our results with the One Health concept:

Page 11, Line 227: “Our results highlight again that our health is closely interconnected with our environment, and exchange of resistance genes among environment, animals and humans may be easier than previous recognized. Thus in dealing with the antibiotic resistance crisis “One Health” approaches, which emphasize collaborative and comparative work from different fields, should be prompted³⁹.”

2. What is the contribution of gene transfer from antibiotic producers to the overall diversity of transferable antibiotic resistance genes in the clinics?

It would be useful for the readers if the authors, based on their understanding of ARG databases could say something more specific about the proportion of r-genes currently known that are likely to originate from antibiotic producers. This paper is important by qualitatively demonstrating sources and a pathway, - the discussion section should include a discussion about the relative importance, from the authors point of view.

This is indeed a very important question. Unfortunately, it is not possible with the currently available datasets to provide a precise estimation of the clinically relevant resistance genes that originate from *Actinobacteria*, since ancient transfers can be very difficult to prove or disprove. Additionally, we would like to refrain from making such estimations since only a minute portion of bacterial genomic diversity (especially for

Actinobacteria) has been characterized which would bias a global analysis. We have updated the manuscript to include the following text referring the readers to two recent review papers that introduced all the possible origins of clinical ARGs.

Page 10, line 205: “Clinical ARGs have complex and diversified origins^{6,16}, and our results confirm that *Actinobacteria* are an important one of them.”

Added references:

6 Wright, G. The Origins of Antibiotic Resistance Vol. 211 *Handbook of Experimental Pharmacology* (ed Anthony R. M. Coates) Ch. 2, 13-30 (Springer Berlin Heidelberg, 2012).

16 Dantas, G. & Sommer, M. O. A. Context matters — the complex interplay between resistome genotypes and resistance phenotypes. *Curr. Opin. Microbiol.* **15**, 577-582 (2012).

3. As pointed out by the authors, wide HGT are perhaps not, from a mechanistic point of view, likely to occur frequently. In addition, there may be other constraints to a protein’s function in a new host.

Both transfer barriers as well as barriers to functionality in new distant hosts are expected to limit stable gene transfers from antibiotic producers. The authors should comment on the latter barrier as well, in particular since efflux pumps are considered in the manuscript, - exemplifying proteins that may depend on other host membrane properties for adequate functioning and regulation.

We thank the reviewer for this suggestion. We have added the following text in the manuscript:

Page 11, line 223: “In addition to being a gene transfer barrier, different phyla also mean distinct cell environments for gene expression, regulation and protein function³⁸. For example, due to the different cell membrane structure in *Proteobacteria*, drug efflux pumps like Cmx and LmrA will export compounds into periplasm instead of directly into extracellular environment as in *Actinobacteria*. Future studies are required to understand how the newly obtained resistance gene and their new hosts will evolve after the gene transfer”

3. Early studies by J. Davies and others showed that antibiotic preparations contained DNA from the commercial production strains - can something be said from this study about the relevance (and impact-if any) of such practices. I.e. its likely that antibiotic preparations have been contaminated with DNA from decades since the 1940s.

We agree that it is an interesting hypothesis. Yet, based on our work, we don't think ARGs encoded by these DNA can be easily transferred to Gram-negative pathogens if it is not linked with pathogens' sequence as in the "carry back" model. However, it won't be surprising if these ARGs are taken up by pathogenic *Actinobacteria*, like *Mycobacterium tuberculosis* and *Corynebacterium diphtheria*, for example, postulated by Pang, Y., et al. (1994). *Antimicrobial Agents and Chemotherapy* 38(6): 1408-1412)

4. Carrier DNA can be a number of mobile genetic elements with broad host range. The need to introduce a new concept of "carriage" could be made clearer.

Here the "carrier sequence" mediates the incorporation of *cmx* into proteobacterial genome by homologous recombination. In this regard, it is different from the classic concept of mobile genetic elements as carrier of ARGs spread, where the transfer depends on the activity of the mobile elements. It has been shown that in natural transformation, homologous recombination is much more efficient than mobile elements mediated recombination (i.e., transposition or integration). Domingues, S. et al. PLoS Path. 8, e1002837 (2012). We improved our text to better emphasise this:

page 8 line 146: "First, a proteobacterial sequence is transferred from *Proteobacteria* into *Actinobacteria* by conjugation, a mechanism known to be highly efficient (Fig. 3a1 and a2)³⁰. Next it recombines with actinobacterial DNA, for example, by the transposition of *cmx* transposon, forming a sandwich structure of actinobacterial DNA flanked by proteobacterial DNA (Fig. 3a3). Then, the sandwich structured DNA released from dead actinobacterial cells is taken up by nearby *Proteobacteria* through natural transformation and incorporated via homologous recombination (Fig. 3a4)."

4. Page 6, line 126, reword "highly identical"

At page 5 line 97 we changed the phrase to "Furthermore, its gene was found to be identical or almost identical (identity over 99%) to genes from many non-*Streptomyces Actinobacteria* (Fig. 2)".

5. Page 11, line 230, reword “Homologous ARGs”

At page 10 line 189 we changed the phrase to “Cmx family proteins were widely spread among *Streptomyces* and other *Actinobacteria* in nature”.

Reviewer #2 (Remarks to the Author):

1. This manuscript presents experimental and bioinformatic evidence for the hypothesis that antibiotic resistance genes have their ultimate origins in antibiotic producing organisms. This hypothesis presupposes that antibiotics themselves might be toxic at all concentrations, and are toxic as soon as they are synthesized within cells. Consequently producers would require resistance mechanisms. Whether this is generally the case is perhaps debatable, but it certainly might be the case in some specific circumstances.

If this were the case, then the authors should refrain from the over-used ‘reservoir’ metaphor, since it implies the genes are somehow in storage, rather than being actively engaged in protecting the host cell.

First, we would like to thank the reviewer for his positive comments regarding our manuscript.

We agree that there are also other explanations about the natural roles of antibiotics/ antibiotic resistance genes. As this background information is rather extensive and outside the main scope of this manuscript, we refer to two review papers on this topic and rephrased the corresponding paragraph to:

page 2 line 34: “In most cases, the biosynthetic gene clusters encoding the biosynthesis of these compounds also contain resistance genes as a self-protecting mechanism towards these compounds² or to modulate their signalling activity³”.

At page 1 line 24: ‘reservoir’ was rephrased as “our results also highlight that the rich resistome harboured by soil *Actinobacteria* can act as a source of new clinically relevant ARGs in modern times”.

2. The manuscript suffers from the somewhat scattered approach to the question, the lack of details around the data that have been assembled, and the strength of the conclusions that can be reached on the basis of these data. The first piece of evidence presented centres around the puromycin resistance gene, *pac*. The text of the paper suggests that the puromycin synthesis gene cluster and the resistance gene *pac* were transferred from *Streptomyces* into *Xenorhabdus maltophilia*, and that subsequently just the *pac* gene was transferred into

Photorhabdus temperata (Figure 2, lines 93-96). This conclusion is not supported by the data. Yes, it is possible that this operon was transferred between Actinobacteria and Proteobacteria at some time, but the significant differences in protein identity suggest that this event happened in the deep evolutionary past. Consequently, one cannot identify the donor or the recipient of the HGT with any certainty,

For the revised version of the manuscript, we updated the database (now including 64,580 genomes and 55,667,859 unique proteins), which is 2.5 times bigger than the previous version and repeated the analysis. We also standardized our analysis protocol as explained in the new **Methods**, and focused only on the discoveries that can be well supported by our data.

We agree with the reviewer that in the example of *pac*, the inter-phylum transfer happened in the deep evolutionary past. Therefore, we now moved this example from the main text to supplementary Fig 2. In this example, we have confidence that the transfer direction was from *Actinobacteria* to *Proteobacteria*, because all the seven proteins including *Pac* found in *Xenorhabdus maltophila* were more similar to actinobacterial proteins than to proteins of any other phyla.

A statement describing this was added as at page 4 line 84:

“In addition to *pac*, the products of six neighbouring genes also show higher similarity to actinobacterial proteins than to proteins from any other phyla, suggesting they were transferred together from *Actinobacteria* to *Proteobacteria* (Supplementary Fig. 2)”.

The phylogenetic trees of these proteins also support this direction, added now in Supplementary Fig. 1. We think that the *pac* example discussed in our manuscript provides much stronger support for the “producer hypothesis” than previously reported examples, where only the similarity of ARG proteins was considered. One pair of proteins showing sequence similarity is not necessary a result of HGT. The similarity can also be resulted from convergent evolution. But in this example seven pairs of proteins all showed similarity. The probability of getting this result by convergent evolution is just too low.

3. and since the HGT event occurred thousands, if not millions of years ago, it does not have much relevance to the current antibiotic resistance crisis. It is also certain that any number of diverse genes have been moved by

HGT between phyla over the millennia, so the observation that some of these might encode antibiotic resistance should not be surprising.

It is indeed necessary to talk about ancient and recent inter-phylum disseminations separately, as they have different relevance to our health. We therefore have modified the whole structure of the manuscript to reflect this and emphasized this aspect explicitly:

page 2 line 46: “Compared with ancient transfers, recent ARG dissemination from *Actinobacteria* to pathogens may pose an even more urgent threat to humans⁹, as *Actinobacteria* ARGs make up a large portion of the environmental resistome”

Although less relevant to the current antibiotic resistance crisis, we think studying the ancient HGT of ARGs is also necessary. It has been proposed that in the ancient HGT of ARGs resistance genes were transferred from *Actinobacteria* to some non-pathogenic *Proteobacteria* likely in soil. Then, in modern times, selected by our use of antibiotic, these resistance genes may have been or can be obtained by pathogenic *Proteobacteria*. Thus, knowing about the processes involved in spreading ARGs may help us predicting new clinical resistance genes. We have added a discussion of this aspect:

page 10 line 206: “Ancient transfers of ARGs from *Actinobacteria* to *Proteobacteria* are proposed to have occurred in soil environment which now can be further acquired by pathogens in modern times³⁵. Recent transfers of *cmx* and *lmrA* from *Actinobacteria* to proteobacterial pathogens suggest that the dissemination may have been accelerated due to the selective pressure of antibiotic use, and raise the alarm that new clinical ARGs can still be generated this way in the future.”

HGT between phyla over the millennia” is not a new idea, but there are still many critical questions to answer, especially regarding HGT of ARGs. For example how many and which ARGs have been transferred, were they transferred to environmental strains or clinical strains, and what mechanism mediated the transfer. In the new Fig.1 we added the information about the antibiotics, resistance mechanisms and clinical relevance of the transferred ARGs. And we proposed that the “carry back” mechanism was also responsible for the ancient HGTs (although it is best supported by the recent HGT samples):

page 11 line 212 “A recent study suggested that the conjugation of *Proteobacteria* to *Actinobacteria* may happen frequently in soil³¹. Thus the “carry back” mechanism may have mediated the ancient HGTs from *Actinobacteria* to *Proteobacteria* in soil using conjugative plasmids as carrier sequence.”

4. The second piece of evidence is based on bioinformatic analyses of antibiotic resistance gene proteins from *Streptomyces*, recovered from the ARDB and CARD databases. These were used to search the Proteobacteria protein database for homologues. Those proteins that generated good hits were then used in further searches and to generate phylogenies (presented as supplementary data). Examining these phylogenies shows diverse patterns of potential inheritance: some genes appear to be restricted to Proteobacteria, others appear to have been acquired from Actinobacteria by HGT, others appear to have been donated to Actinobacteria from Proteobacteria, and in other cases the phylogenies suggest extensive movement to diverse phyla by HGT. So the conclusions drawn here suffer from confirmation bias. Yes, there are instances where the most parsimonious explanation is HGT from Actinobacteria, but there are also examples of every other kind of HGT movement and direction. To concentrate just on the Actinobacterial to Proteobacterial movement as if this tells us something novel is to miss the point entirely.

As stated in our reply to comment 2, for the revised version, we updated the sequence database (from 23,495 genomes to 64,580 genomes), which resulted in improved phylogenetic trees (supplementary Fig 1). We focused only on the examples that transfer and direction can be well supported by the data. These are shown in the new Figure 1.

We are not trying to say all *Streptomyces* ARGs have disseminated to *Proteobacteria*, nor the HGTs can happen in only one direction. In our studies, we indeed also detected ARG transfers from *Proteobacteria* to *Actinobacteria*. Our proposed ‘carry back’ mechanism relies on bi-directional horizontal gene transfer, since the first step is the transfer of DNA sequence from *Proteobacteria* to *Actinobacteria* by conjugation. For example in the ‘carry back’ of *cmx*, a resistance gene *sul1* was transferred from *Proteobacteria* to *Actinobacteria*.

We have updated Supplementary Figure 1 and added the new data accordingly:

page 4, line 78. ‘‘HGTs from *Proteobacteria* to *Actinobacteria* were also noticed in some trees, showing that the gene communication between these two phyla could be bidirectional (Supplementary Fig. 1). For example one of the *Streptomyces* ARG *sul1* (AFN41071.1) is likely to be recently acquired from *Proteobacteria* with high sequence identity of 95%.’’

5. Finally, the authors investigate the possibility that mobile DNA elements with their origins in Proteobacteria can infiltrate the Actinobacterial genome and shuttle genetic material back to the Proteobacteria. They have convincingly demonstrated this in laboratory assays, backed up with some bioinformatic observations. Again, this is not surprising, and the focus here should be on the fitness of mosaic mobile elements whose evolutionary success lies in their ability to associate with genes that confer advantageous phenotypes. The present manuscript reads somewhat teleologically, as if this is a strategy specifically employed by the Proteobacteria.

To the best of our knowledge, we are not aware of any publication describing recent ARG transfer from *Actinobacteria* to *Proteobacteria* or postulation of actual mechanisms of transfer, thus the referral of antibiotics producing bacteria as source of clinical relevant ARGs has always been speculative. We are not aware of any theory similar to our “carry back” model that has been reported and experimentally supported before (von Wintersdorff, C. J. H., et al. 2016. *Frontiers in Microbiology* **7**: 173.). People do know that *Proteobacteria* can conjugate with *Actinobacteria*, but little is known about the fate of the transferred sequence in nature. No study has been published about if the transferred sequence can carry exogenous sequence back to *Proteobacteria*. And even though the theoretical possibility exists, there has not been actual example reported. So our finding for the first time confirmed that the inter-phylum transfer actually has an efficient mechanism rather than just by random illegal recombination which has an extremely low efficiency. To emphasize this in the manuscript, we have added a paragraph:

page 6 line 111. “Conjugation or transduction from *Actinobacteria* to *Proteobacteria* has yet to be known. Natural transformation of *Proteobacteria* is able to take up free DNA, but due to the low sequence homology, actinobacterial DNA has a low probability of being incorporated into proteobacterial genomes by homologous recombination²⁵. In theory, the DNA can be inserted randomly by non-homologous end joining, but the frequency is extremely low²⁶”.

We agree with the reviewer that mobile elements have an important role in the “carry back” model. The “carry back” relies on both the conjugative sequence and the acceptor bacteria (natural competence). And from a viewpoint of evolution, “carry back” phenomenon benefits both the mobile elements and the acceptor bacteria. To avoid the “teleologically”, we deleted “*Proteobacteria* play an initiating role because of their ability

to conjugate across phyla and take up free DNA... the “carry back” model might be a general mechanism to acquire new genetic information from surrounding organisms and not limited to *Actinobacteria*”.

Mosaic mobile elements are ideal carrier sequence. However, in the “carry back” model their role as a carrier is different from their classic roles. For example, as a result of the carry back of *cmx*, *cmx* was incorporated into conjugational class I integron in *Proteobacteria*. This was actually achieved not by the activity of the integrase but by homologous recombination, and the DNA entered into *Proteobacteria* cell actually not by conjugation but by natural transformation. Therefore we think the “carry back” mechanism should be discussed separately from the class concept of mosaic mobile elements. We think that “carry back” is also possible with just simple conjugational plasmids and responsible for the ancient ARG transfers before mobile multidrug resistant unites (mosaic mobile elements) had evolved. Of course in modern times, the “carry back” is further facilitated by the mosaic mobile elements which are more powerful than simple conjugative plasmids.

We discuss this page 11 line 211:

“*Proteobacteria* are well known to be able to conjugate with organisms from other phyla and even other kingdoms³⁴, and recent study suggested that the conjugation of *Proteobacteria* with *Actinobacteria* may happen frequently in soil³⁰. Thus, the “carry back” mechanism may have mediated the HGTs from *Actinobacteria* to *Proteobacteria* in soil using conjugative plasmids as carrier sequence. In modern times, because of our use of antibiotics, mobile genetic elements including conjugative plasmids, integrons and transposons tend to be clustered together with ARGs forming mobile multidrug resistant unites which showed extraordinary capability of spreading among commensals, pathogens and even environmental bacteria in water and soil³⁷. Not surprisingly the carrier sequence of *cmx* was from a widespread conjugative class I integron and is composed of transposon *IS6100* and resistance gene *sulI*. Accordingly, *cmx* carried back was incorporated into such unites making it immediately ready for further dissemination”.

We understand that recent ARG dissemination is of the most relevance to our current drug resistance crisis. In addition to that, the “producer theory” is also a fundamental question in the fields of natural products, bacterial ecology and bacterial evolution. So we think it is justified to include in this paper the occurrence of “carry back” in ancient time and in soil environment as well.

6. The manuscript also requires significant copy editing, both to correct a number of typographical errors, and to improve the English expression. The methods and processes employed during the work need to be explained in more detail, and the rationale behind those procedural decisions needs to be justified.

The manuscript was edited by two native English speakers to help us with the language, and the manuscript has been rewritten according to the reviewer's comments.

Reviewer #3 (Remarks to the Author):

1. The authors purport evidence of a new mechanism for horizontal gene transfer between distantly related bacteria. Actinobacteria have long been appreciated to be the source of many of the clinically relevant antibiotics used today. As expected, and well documented in previous literature, these bacteria must also encode resistance determinant to the antibiotics they produce. As documented in many publications, this provides a wealth of antibiotic resistance determinants in the natural environment. The current manuscript does not provide any new information in this area.

We thank the reviewer for the comments, which were very helpful for improving the manuscript. However we do not agree with the reviewer that “the current manuscript does not provide any new information in this area.”

To the best of our knowledge, this hypothesis has been proposed for only a few resistance genes (summarized in table 1). The hypothesis originated from the observation that the mechanism of resistance found in pathogens and those found in antibiotic producers is the same and that resistance proteins from pathogens and antibiotics are homologous. The “producer hypothesis” provides a logical explanation for the observed similarities, but by any standard, it is not scientifically convincing to claim an origin relationship just based on the protein sequence identity of 32-64% (as a reference, the RNA polymerases sequence identity is 58% between *Streptomyces* and *E.coli*).

Table 1: clinical resistance genes that have been proposed to have originated from *Actinobacteria*:

ARG	Sequence identity	comments
aminoglycoside antibiotic-inactivating enzymes ¹	51%	also supported by our data
β -lactamase ²	52%	controversially discussed ³
Erythromycin resistance rRNA methyltransferases ⁴	32%	ARG of Gram-positive pathogens, not within the scope of this article.
vancomycin resistance VanHAX ⁴	61-64%	ARG of Gram-positive pathogens, not within the scope of this article.

Although the “producer origin hypothesis” has been stated half a century ago it remained a hypothesis to be confirmed. This is well reflected in many papers:

Forsberg, K. J., et al. (2012). "The Shared Antibiotic Resistome of Soil Bacteria and Human Pathogens." *Science* **337**(6098): 1107-1111. :

“Soil, one of the largest and most diverse microbial habitats on earth, is increasingly recognized as a vast repository of antibiotic resistance genes (9–13). Not only does soil come into direct contact with antibiotics used extensively in rearing livestock (14) and plant agriculture (15), but it is also a natural habitat for the Actinomycete genus Streptomyces, whose species account for the majority of all naturally produced antibiotics (16). Despite numerous studies demonstrating that soil contains resistance genes with biochemical mechanisms similar to those in common pathogens (3, 11–13), the sequence identities of these genes diverge from those of pathogens (17), providing little evidence that these resistomes have more than an evolutionary relationship. Therefore, whether soil has recently contributed to or acquired resistance genes from the pathogenic resistome remains an open question, and accordingly, the role of soil in the current global exchange of antibiotic resistance remains poorly defined.”

Dantas, G. and M. O. A. Sommer (2012). "Context matters — the complex interplay between resistome genotypes and resistance phenotypes." *Current Opinion in Microbiology* **15**(5): 577-582. :

"Of note, while antibiotic producer soil Actinomycetes have been clearly demonstrated to be phenotypically multidrug resistant [29,30] and to express resistance proteins with very similar mechanisms to those found in pathogens [26], their resistance genes have thus far been found to be highly sequence divergent from those of pathogens [48]. Therefore, the 'producer hypothesis' question, namely whether the resistome of antibiotic producers are in current exchange with the resistome of clinical pathogens, remains unresolved."

For the first time, our work now provides in silico and experimental data to proof this "old" hypothesis:

- (1) We found 100% DNA sequence identity between actinobacterial ARG and proteobacterial pathogen ARG.
- (2) We found the mobile elements and mechanism carrying the ARG transfer.
- (3) We provide direct experimental proof for that resistance genes can be transferred from *Actinobacteria* to Gram-negative bacteria, whereas no such experimental data existed previously.

We have rewritten our Introduction to clarify this:

page 2 line 36: " As early as 1973 it was hypothesized that the aminoglycoside antibiotic-inactivating enzymes found in Gram-negative pathogens could have originated from the ARGs of actinobacterial antibiotic producers through ancient horizontal gene transfer (HGT), based on the discovery that they employ the same enzymatic mechanisms⁴. Afterwards this "producer origin hypothesis" was also proposed for some class A β -lactamases⁵ of Gram-negative pathogens and erythromycin⁶ and vancomycin resistance genes⁷ of Gram-positive pathogens. However, in these cases the sequence similarities of ARG proteins between *Actinobacteria* and Gram-negative pathogens are low, making it difficult to distinguish if they are really resulted from ancient HGT or rather from other kinds of evolutionary processes⁸. And no DNA sequence evidence was left to study their possible transfer mechanisms."

Many metagenomic studies did confirm "a wealth of antibiotic resistance determinants" in soil *Actinobacteria*. This is not surprising. The key question here is whether these ARGs can threat our health by being

disseminated to human pathogens. To our best knowledge, not a single case of direct link (i.e., recent dissemination evidenced by DNA sequence similarity) between actinobacterial resistome and clinical resistome has been reported. Also, no HGT mechanism for the gene transfer from *Actinobacteria* to *Proteobacteria* has been described so far. Thus, it is not clear if the extensive resistome of *Actinobacteria* in fact has contributed to resistance in human pathogens. In contrary, several recent studies reported finding no trace of such inter-phylum transfer of ARG in their resistome surveys and emphasized that inter-phylum HGT should be very difficult:

- Pehrsson, E. C. et al. Interconnected microbiomes and resistomes in low-income human habitats. *Nature* 533, 212-216 (2016).
- Forsberg, K. J. et al. Bacterial phylogeny structures soil resistomes across habitats. *Nature* 509, 612-616 (2014).
- Sommer, M. O. Microbiology: Barriers to the spread of resistance. *Nature* 509, 567-568 (2014).

Our paper for the first time confirmed the direct interconnection between the actinobacterial resistome and the clinical resistome. We have further emphasized this point in the revised manuscript, and want to make the scientific community aware of that such ARG dissemination is quite active in the timescale of our antibiotic resistance crisis era, rather than just occasionally happened in the long evolutionary history. We rephrased our language as:

page 2 line 46 “Compared with ancient transfers, recent ARG dissemination from *Actinobacteria* to pathogens may pose an even more urgent threat to humans⁹, as *Actinobacteria* ARGs make up a large portion of the environmental resistome. In addition to self-protecting ARGs, most *Actinobacteria* strains also carry ARGs horizontally obtained from other *Actinobacteria*^{6,10}. In an investigation of actinomycetes from soil, isolates were on average resistant toward seven to eight antibiotics from a collection of 21 representative antibiotics¹¹. In a functional metagenomics study using *Escherichia coli* as expression host, *Actinobacteria* was found to be the most enriched source of resistance-conferring DNA fragments relative to their abundance¹². However, since no recent transfer from *Actinobacteria* to Gram-negative pathogens has been discovered and many recent studies showed that phylogenetic and ecological boundaries are two major ARG transfer barriers¹²⁻¹⁵, the clinical relevance of this large resistome remains unclear^{16,17}.”

2. A second claim in the manuscript is that a direct natural source of transfer between members of the phylum Actinobacteria and proteobacteria has been identified. However, the actual evidence reported to support this idea involves using laboratory cloning procedures to make a variety of recombination intermediates that could have occurred via transfer through various intermediate bacteria. Therefore, no new information illuminating a hereto unknown recombination step is provided in this work. In summation, the current work simply summarizes previous well established findings without providing an new mechanisms of transfer between distant important phyla.

The only artificial construct in this experiment is the cloned carrier sequence. We used an *Acinetobacter baylyi* strain with a cloned carrier sequence to resemble natural Gram negative bacteria with the same sequence. Several pathogenic *Acinetobacter baumannii* strains (as shown in our table S5) naturally harbor this sequence, and they are also naturally transformable. Studies have shown that *Acinetobacter baumannii* and *Acinetobacter baylyi* behave quite similarly in many aspects including natural transformation. (Peleg, A. Y., et al. 2008. Clinical Microbiology Reviews 21: 538-582). We used *Acinetobacter baylyi* because it is nonpathogenic, and it has been used in many previous studies as a model to study HGT of ARGs among *Proteobacteria*, or from GE plants to *Proteobacteria*^{6,7}, also because this way it is easier to have a negative control without the carrier sequence. We also repeated the experiment using killed donor cells instead of purified DNA to better mimic the natural process. In the revised version of the manuscript we write:

page 9 line 172: “To provide further evidence for the hypothesis, we reproduced the inter-phylum gene transfer of *cmx* experimentally. *C. resistens* DSM 45100 which naturally has the sandwich structure sequence was used as the actinobacterial *cmx* donor. *Acinetobacter baylyi* ADP1, a model strain for HGT studies^{27,34} (Fig. 3c) with a cloned IS6100-orf5-sulI sequence was used as the receptor to mimic proteobacterial pathogens with the same sequence, such as *Acinetobacter baumannii* D4 (supplementary table 4). Heat-killed and lysed *C. resistens* cells were added into *A. baylyi* ADP1 culture, incubated and followed by selection on kanamycin plate”

DNA transfers by conjugation from Proteobacteria to Actinobacteria and transposition of *cmx* transposon inside Actinobacteria have been already confirmed in previous studies:

At page 11 line 211 “*Proteobacteria* are well known to be able to transfer DNA to organisms from other phyla and even other kingdoms by conjugation³⁴. A recent study suggested that the conjugation of *Proteobacteria* to *Actinobacteria* may happen frequently in soil³⁰”

At page 6 line 122 “we examined if the transposition activity, which has been experimentally validated in *Corynebacterium glutamicum*²⁹”

Therefore, we did not try to repeat them in this experiment, instead we used a *C. resistens* DSM 45100 directly to be the *cmx* donor. *C. resistens* DSM 45100 naturally has a sandwich structure. It is not constructed by artificial cloning.

To the best of our knowledge there is no other similar idea reported before. Most importantly, not even a good hypothesis about how the inter-phylum HGT could have happened to explain the “producer hypothesis” was published so far. Although very convincing, the hypothesis was based on assumptions that it may have happened somehow (for example through illegal recombination or non-homologous end joining) at an extremely low frequency in the million years of evolutionary history. In our study, we could experimentally demonstrate that the “carry back” mechanism is highly efficient.

This aspect was further stressed out in the Discussion section at page 11 line 195ff.

REVIEWERS' COMMENTS:

Reviewer #1 (Remarks to the Author):

The clarity and quality of the manuscript has been improved by taking into account the many constructive suggestions made by the reviewers.

Possible fitness effects of HGT events are not considered but appears to be outside the scope of this particular study.

Reviewer #2 (Remarks to the Author):

The authors have addressed all my comments, and although I am still not entirely convinced by the data and the conclusions drawn, the ideas and observations are novel and thought provoking.

The manuscript still requires considerable editing and correction throughout. There are numerous grammatical and spelling errors even in the newly prepared sections included in the response to reviewers.

I understand that the authors have has the MS read by English speakers, but it might be necessary to have a professional editing service polish the writing.

REVIEWERS' COMMENTS:

Reviewer #1 (Remarks to the Author):

The clarity and quality of the manuscript has been improved by taking into account the many constructive suggestions made by the reviewers.

Possible fitness effects of HGT events are not considered but appears to be outside the scope of this particular study.

We thank the reviewer very much for the positive comments regarding our manuscript.

Reviewer #2 (Remarks to the Author):

The authors have addressed all my comments, and although I am still not entirely convinced by the data and the conclusions drawn, the ideas and observations are novel and thought provoking.

We thank the reviewer very much for the positive comments regarding our manuscript. We have removed the overstatements and speculations as suggested. Some of them will be tested in our following studies.

The manuscript still requires considerable editing and correction throughout. There are numerous grammatical and spelling errors even in the newly prepared sections included in the response to reviewers.

I understand that the authors have has the MS read by English speakers, but it might be necessary to have a professional editing service polish the writing.

Thanks for the suggestion. Nat. Commun. has kindly helped us with the language throughout the manuscript.